# AP-2δ Is the Most Relevant Target of AP-2 Family-Focused Cancer Therapy and Affects Genome Organization

**DOI:** 10.3390/cells11244124

**Published:** 2022-12-19

**Authors:** Damian Kołat, Lin-Yong Zhao, Mateusz Kciuk, Elżbieta Płuciennik, Żaneta Kałuzińska-Kołat

**Affiliations:** 1Department of Experimental Surgery, Medical University of Lodz, 90-136 Lodz, Poland; 2Gastric Cancer Center and Laboratory of Gastric Cancer, State Key Laboratory of Biotherapy, West China Hospital, Sichuan University, and Collaborative Innovation Centre for Biotherapy, Chengdu 610041, China; 3Department of Molecular Biotechnology and Genetics, University of Lodz, 90-237 Lodz, Poland; 4Doctoral School of Exact and Natural Sciences, University of Lodz, 90-237 Lodz, Poland; 5Department of Functional Genomics, Medical University of Lodz, 90-752 Lodz, Poland

**Keywords:** transcription factors, targeted therapy, ligandability, cancer, genome organization, bioinformatics, AP-2, TFAP2, AP-2δ, TFAP2D

## Abstract

Formerly hailed as “undruggable” proteins, transcription factors (TFs) are now under investigation for targeted therapy. In cancer, this may alter, inter alia, immune evasion or replicative immortality, which are implicated in genome organization, a process that accompanies multi-step tumorigenesis and which frequently develops in a non-random manner. Still, targeting-related research on some TFs is scarce, e.g., among AP-2 proteins, which are known for their altered functionality in cancer and prognostic importance. Using public repositories, bioinformatics tools, and RNA-seq data, the present study examined the ligandability of all AP-2 members, selecting the best one, which was investigated in terms of mutations, targets, co-activators, correlated genes, and impact on genome organization. AP-2 proteins were found to have the conserved “TF_AP-2” domain, but manifested different binding characteristics and evolution. Among them, AP-2δ has not only the highest number of post-translational modifications and extended strands but also contains a specific histidine-rich region and cleft that can receive a ligand. Uterine, colon, lung, and stomach tumors are most susceptible to AP-2δ mutations, which also co-depend with cancer hallmark genes and drug targets. Considering AP-2δ targets, some of them were located proximally in the spatial genome or served as co-factors of the genes regulated by AP-2δ. Correlation and functional analyses suggested that AP-2δ affects various processes, including genome organization, via its targets; this has been eventually verified in lung adenocarcinoma using expression and immunohistochemistry data of chromosomal conformation-related genes. In conclusion, AP-2δ affects chromosomal conformation and is the most appropriate target for cancer therapy focused on the AP-2 family.

## 1. Introduction

The concept of Transcription Factors (TFs) targeting is becoming increasingly popular [1]. Formerly regarded as “undruggable” DNA-binding proteins, they are now investigated in studies examining their activity depending on the use of various selective modulators [2,3]. Currently available approaches include targeting nuclear hormone receptors or protein–protein interactions, modulating proteasomal TFs degradation or TF’s drivers, degrading TFs with PROteolysis TArgeting Chimeras (PROTACs), modifying TFs post-translationally, or disrupting TF-DNA binding [4,5,6,7,8,9,10,11]. In cancer, these interventions may not only increase cell death but block the immune evasion, Epithelial-to-Mesenchymal Transition (EMT), self-renewal, treatment resistance, replicative immortality or autoregulatory cancer-driving circuits [12]. Most of these were found to be implicated in genome reorganization [13,14,15,16], the process that accompanies a multi-step tumorigenesis and which frequently develops in a non-random manner [17]. In the current post-genomics era, three-dimensional genomics is a frontier field and is primarily focused on the relationship between chromatin spatial conformation and regulation of gene transcription [18]. It is estimated that one-fifth of the so far identified oncogenes are in fact TFs [19]. Still, targeting-related research on some TF families is scarce, for example, within Activating enhancer-binding Protein 2 (AP-2); this is unexpected because the members of this family are known for their altered functionality in cancer and prognostic importance in oncological patients [20,21]. Typically, AP-2 transcription factors are expressed during embryogenesis and orchestrate the cell cycle, proliferation, and apoptosis, enabling appropriate development of, inter alia, limbs, eyes, or facial features [22,23]. Apart from our previous study that paved the way for AP-2α and AP-2γ to be considered as targeted therapy candidates [24], other authors [19] have put a different family member, AP-2δ, on the list of nearly three hundred oncogenic transcription factors and regulators, which could be a useful hint for TF-based cancer treatment. However, no studies were focused on the structural properties that would justify the rationale for targeting AP-2 proteins or could describe how to perform it selectively, without targeting other AP-2 family members. Moreover, no inhibitors are currently known for AP-2 factors, complicating the research progress. In contrast to many previous experiments, which do not consider all five (α, β, γ, δ, ε) representatives of AP-2 family at once (even our previous research focused on the two best-described ones), the present study examines the relevance of each AP-2 member to be selectively recognized; it then identifies the best AP-2 protein from the point-of-view of selective ligandability and provides related data about its mutations, genetic targets, transcriptional co-factors and correlated genes in a pan-cancer spectrum. A specific stage of research was also dedicated to genes that govern chromosome conformation, in order to propose future directions in the AP-2 context since the interplay between this family and chromosomal conformation is insufficiently researched, while transcription on its own can shape the spatial genome structure [25].

## 2. Results and Discussion

### 2.1. AP-2 Proteins Have Conserved the “TF_AP-2” Domain, but the N-Terminal Amino Acid-Rich Region of AP-2δ Is Unique

At first, AP-2 transcription factors were compared evolutionarily, functionally, and both in terms of how they bind to DNA sequences and are bound by the interacting proteins. Typically, AP-2 factors interact with other proteins with the use of a proline-rich motif (PPxY) while the cooperation within the AP-2 family is maintained via a so-called “TF_AP-2” domain that is responsible for dimerization (of homo- and hetero- dimers), as well as DNA binding (this region contains Helix-Span-Helix [HSH] motif) [26]. Clustal Omega revealed that a part of the sequence corresponding to the amino acid-rich motif has the pattern “PPPY” in AP-2α, AP-2β, AP-2γ, and AP-2ε, compared to “STNH” in AP-2δ (Figure 1A). Moreover, the DNA sequences that are recognized by AP-2 factors are also distinct (Figure 1B). While the length of the consensus logo for AP-2δ was congruent to that of AP-2α and AP-2γ, the Jaccard index suggested a slightly greater similarity of content between AP-2δ and AP-2β than between AP-2δ and AP-2α/AP-2γ. No data is available for AP-2ε based on HOmo sapiens COmprehensive MOdel COllection database (HOCOMOCO). Furthermore, the phylogenetic tree indicates that the AP-2δ evolved separately from the rest of the AP-2 family (Figure 1C). However, since the “TF_AP-2” domain is highly conserved [23] and encompasses the rest of the protein sequence in the direction of carboxyl terminus (Figure 1D), their distinction is supposedly due to differences located in the “first half” of the polypeptide chain, i.e., from amino terminus to the middle part of the amino acid sequence.

Although the last 20 years did not yield many findings related to the role of AP-2δ in carcinogenesis (this TF was cloned in 2001 [27], and the research focused more on embryogenesis [28,29,30,31]), some interesting observations were noted, e.g., its distinct binding affinity to conserved AP-2–binding sites [27]. This suggests that AP-2δ might regulate genes in vivo through other mechanisms than the remaining AP-2 TFs, presumably via interaction with unique coactivators. It has also been proposed that AP-2δ might function as a negative regulator, inhibiting or modulating the transactivation capability or DNA-binding affinity of the other AP-2 family members [23]. What also makes AP-2δ unique is the more restricted temporal and spatial expression pattern when compared to other AP-2 factors–among normal tissues, the expression of this TF was observed mainly in the central nervous system, retina, and developing heart [28]. AP-2δ is encoded by Transcription Factor AP-2 Delta (*TFAP2D*) gene, which shares the same cytological location (chromosome 6p12.3) with *TFAP2B* (encoding AP-2β) [26]. This explains the previous symbol of *TFAP2D*, originally denoted as *TFAP2BL1* (“BL1” stands for “beta-like 1”) [32]. Interestingly, a human-specific Alu DNA cassette was found flanking the *TFAP2D* and *TFAP2B* genes, which may affect their regulation [33]—this leaves the investigational window open, especially since Alu elements have a role both in cancer and genomic rearrangements [34,35].

### 2.2. AP-2δ Has the Least Similar Protein Sequence That Contains the Highest Number of PTM Sites and Extended Strands

Similarities and differences in protein structures of all AP-2 factors were then investigated. Clearly, alpha-helices (that mainly correspond to the “TF_AP-2” domain and a part responsible for dimerization) are highly conserved in the entire family. However, the percentage of the shared identity of all AP-2 factors at once is about 30%, as investigated through Clustal Omega (Figure 2A). Due to the separate evolution of AP-2δ, it was also decided to consider this protein as a reference in pairwise comparisons—this revealed that AP-2δ is only almost half identical to other AP-2 factors. The percentages of identity for all comparisons are summarized in Appendix A. The analysis of the amino acid sequence revealed that out of the entire family, AP-2δ possesses the highest number of Post-Translational Modifications (PTMs), which suggest vast regulation possibilities (Figure 2B). Targeting post-translation modifications is one of the TF-based therapy strategies [36,37]: more post-translational modification sites entail more therapeutic possibilities. Referring to the aforementioned “PPPY” and “STNH” patterns, the UniProt, ProSite, InterPro, and Motif Scan confirmed that AP-2α, AP-2β, AP-2γ, and AP-2ε have proline-rich motifs in this area. However, AP-2δ was the only family member lacking this motif, as it was not identified in any database (Table 1). The dissimilarity in the length of the sequence between UniProt + InterPro versus ProSite + Motif Scan is dictated by the fact that the first two databases indicate the motif, while the latter two demonstrate a wider profile of the region. Nevertheless, data from UniProt and InterPro are consistent with ProSite and Motif Scan: the amino acids of the motif exist within the wider area of the region. Subsequently, it was verified whether the lack of a proline-rich sequence in AP-2δ is compensated by another region that is rich in a specific amino acid. According to ProSite, it turned out the proline-rich region (with an additional alanine-rich region in AP-2ε) that is present in AP-2α/β/γ/ε, overlaps with the histidine-rich region of AP-2δ (Figure 2C). This region was depicted in advance so that all comparable motifs/regions and domains were present in a single graph (Figure 2A). The AP-2δ protein sequence was scanned for histidine, which revealed that almost half of histidines (10 out of 25) are located in the histidine-rich region (Figure 2D). Interestingly, this region includes three patterns where two histidines are in a row (“HH”); this would enable the histidine-rich region to be recognized selectively. For comparison (Appendix A), there is only one identical histidine pattern in AP-2β and one in AP-2γ (AP-2α and AP-2ε do not contain “HH” patterns in their sequences); the consideration of at least two such patterns would allow exclusive recognition of AP-2δ within the AP-2 family. The remaining differences investigated at this stage included the prediction of secondary structures that could be formed in the AP-2 factors. As revealed using the fourth Garnier–Osguthorpe–Robson algorithm (GOR4), out of the entire AP-2 family, AP-2δ has the largest number of extended strands but the smallest of random coils (Figure 2E). Bang et al. suggested that binding affinity could be elevated with the increase of extended strands in secondary structure [38]. Particular attention was also given to the amino acids preceding the conserved “TF_AP-2” domain; these were evaluated for their similarity, physicochemical properties, and structural characteristics (Figure 3). In brief, AP-2δ was found to be the least similar to other AP-2 TFs, which certifies the comparison of the entire protein sequences. Compared to the rest of family, the “first half” of AP-2δ polypeptide chain is more hydrophilic, contains a lower amount of conformationally special amino acids (proline, glycine) but comprises additional cysteines and threonines. Although the regions that mainly contribute to the binding affinity are hydrophobic, hydrophilicity may facilitate drug specificity [39]. The occurrence of cysteines can also reduce the off-target effect and toxicity [40]. From the structural point of view, there is an increment of amino acids having the highest strand propensity, whereas the opposite is found regarding the number of residues with a high ability to form a turn. From the drug-targeting perspective, AP-2δ was also investigated for the content of leucine (L), isoleucine (I), phenylalanine (F), and methionine (M), seeing that the recent study by Wang et al. indicated that drugs frequently bind to these amino acids [41]. Analyzing the entire protein sequences (acquired from the UniProt database), AP-2δ contains the largest sum of “LIFM” amino acids, whereas in the “first half” it is surpassed only by AP-2ε. Collectively, it appears that some properties might be of use with regard to ligand development.

### 2.3. AP-2δ Contains the Most Relevant Ligandability-Related Cleft That Can Be Availed Using a Unique Histidine-Rich Region

The above findings suggest that AP-2δ could be selectively recognized by a potential ligand, at least among AP-2 family members. It was then evaluated whether any relevant clefts are present in the protein structure. The cleft analysis performed using a database containing structural summaries of Protein Data Bank entries (PDBsum) indicated that AP-2δ has the largest cleft out of the entire family; this pocket is twice as large (volume: 15665 Ångströms) compared to the second largest cleft (volume: 7734 Ångströms) present in AP-2β (Figure 4). It is known that ligands preferably bind to larger clefts [42]; moreover, the so-called R1 ratio of the cleft should ideally be >2 [43]. Regarding the biggest cleft of AP-2δ, it has R1 ratio close to 4; the other AP-2 factors have their largest pockets with an R1 ratio not exceeding 2.5, except for AP-2α which has no clefts exceeding 2.

Based on these findings, the three-dimensional structure of AP-2δ was analyzed. At first glance, the presence of a histidine-rich region (in particular: three “HH” patterns) instead of a proline-rich region appears to be a good solution that allows selective identification of AP-2δ among other AP-2 proteins. It is theoretically possible to consider amino acid-rich motifs as therapeutic targets, as has been found previously [44,45]. Alternatively, these histidines could be used as a docking site so that the potential ligand could also recognize a different, nearby sequence. For example, the second-best cleft of AP-2δ is close to histidines 106–107 (H106–107) and there is a space between them (Figure 5A,B). The cleft itself is also composed of histidines (Figure 4D) but not the ones comprising a histidine-rich region; the occurrence of such amino acids in cavities can strengthen the binding [46].

However, many other proteins (outside the AP-2 family) could contain identical “HH” pattern(s); therefore, the ideal solution should be related to the simultaneous recognition of double histidines and the “TF_AP-2” domain (that is exclusive for AP-2 family representatives). Recognition of the “TF_AP-2” domain would exclude proteins that are not representatives of the AP-2 family, whereas recognition of at least two “HH” patterns would exclude other AP-2 factors than AP-2δ. In the entire AP-2δ sequence, there are three “HH” patterns, all within the histidine-rich motif, making it possible to bind to the specific location. It is also worth underlining that the best three clefts in AP-2δ mainly overlap with the “TF_AP-2” domain (Figure 4). Hence, this could be a method that would not necessarily aim at the largest best cleft only, but could act on some other cavities. Since the “TF_AP-2” domain also includes a sequence that is responsible for dimerization, the ligand binding, i.a., on the site of the “TF_AP-2” domain would act as a steric hindrance, inhibiting the function of transcription factor dimers. Hypothetically, there is a space for the ligand to recognize not only the “HH” patterns of the histidine-rich region but also some clefts (including the largest one) that overlap with the “TF_AP-2” domain (Figure 5C,D). Thus, one of the examples showing the rationality for further research in this field is presented. Unfortunately, the research methodology that needs to be performed requires the expensive verification and time-consuming description of much more detailed strategies, most probably via Computer-Aided Drug Design (CADD) and further pre-clinical molecule selection.

### 2.4. The Highest Number of Samples with TFAP2D Mutation Was Observed in UCEC, COAD, LUSC, LUAD, and STAD, the Tumors in Which Cancer-Related Genes Co-Depended with TFAP2D

Taking a pan-cancer view of AP-2δ, the number of samples with the *TFAP2D* gene mutation was determined across tumor types. The Tumor Immune Estimation Resource (TIMER 2.0) indicated that the largest percentage of samples with the mutation was in uterine corpus endometrial carcinoma (Figure 6A), which is in agreement with the FABRIC Cancer Portal (Figure 6B). Collectively, the following six tumors have the highest percentages of samples with *TFAP2D* mutation: Uterine Corpus Endometrial Carcinoma (UCEC), Colon Adenocarcinoma (COAD), Lung Squamous Cell Carcinoma (LUSC), Lung Adenocarcinoma (LUAD), Skin Cutaneous Melanoma (SKCM), and Stomach Adenocarcinoma (STAD). However, while the mutation frequency is not statistically significant in any tumor or even in a pan-cancer view (according to FABRIC), muTarget indicates that several cancer hallmark genes and a few Food and Drug Administration (FDA)-approved drug targets demonstrate significantly different expression between *TFAP2D*-mutant and *TFAP2D*-wild samples of uterine, colon, lung, and stomach cancers (Figure 7A). Regarding cancer hallmark genes for colon, stomach and uterine cancers, only the two best up- and downregulated examples are presented in the figure; the same applies to uterine cancer in terms of FDA-approved drug targets. Appendix A summarizes the results from muTarget, which in addition to cancer hallmark genes and FDA-approved drug targets, includes data for all significant genes identified using this tool. Separately for upregulated and downregulated *TFAP2D* mutation-related genes, the functional analysis was performed among cohorts visible in Figure 7A. The gene ontology revealed that the “cell adhesion” and “cellular protein metabolic process” were the most frequently annotated processes, i.e., found in at least half of functional annotations. It is important to note that the tumors identified through muTarget overlap with those of TIMER/FABRIC, and that some genes from muTarget are predicted genetic targets of AP-2δ (marked with a blue frame). Regarding AP-2δ gene targets, their location in the genome was additionally visualized using Ideogram Viewer and split into predicted and validated ones (Figure 7B,C, respectively). Due to the lack of literature on AP-2δ, it is challenging to find data that can be referenced, not only in specific tumor but also in pan-cancer view. Thus, it is unknown whether some of these observations are biologically more inquisitive than the others. Especially since these genes include representatives of regulation of chemotaxis (*CXCR4*, *CXCL14*, *CXCL5* [47]), DNA replication (*MCM2*, *MCM5* [48]), Extracellular Matrix (ECM) degradation (*MMP2* [49]), transcription (*MYB*, *MYCL*, *ZHX2* [50,51,52]) telomere function (ACD [53]), catalysis (RRM1, DHFR [54,55]), or signaling (*TNFRSF10C*, *IGFBP7*, *FGFR2*, *PRKAB1* [56,57,58,59]).

### 2.5. Some AP-2δ Targets Might Be Located in Adjacent TADs or Act as TcoFs

The experimentally validated AP-2δ targets were investigated for their location on chromosomes, taking into account the three-dimensional genome organization [60] (Figure 8A and Appendix A). The intention was to identify any informative insights into AP-2δ regulation, as currently there is only one Chromatin Immunoprecipitation (ChIP) method (specifically: ChIP-exo [exonuclease]) which was performed as a part of enormous and valuable research [61], but unfortunately provides a small amount of data when it comes to AP-2δ. Our WashU Browser-based investigation (Figure 8B) presented that AP-2δ targets might not only be located in close proximity on a single chromosome (e.g., *TRIM22* and *FANCD* on chromosome 11) but also on different chromosomes (*IFI44* and *MIS12* on chromosomes 1 and 17, respectively). For the chromosomal location of each gene, see Appendix A. Particularly interesting was the example of *MVK* and *VPS29*, in which their genomic locations were found to be the closest to each other among the presented genes. Such close proximity suggested that they may constitute a single Topologically Associating Domain (TAD) [62]; however, as investigated via TAD Knowledge Base (TADKB), they are a part of two adjacent TADs (Figure 8C and Appendix A). Although DNA sequences of single TAD interact more frequently than those from neighboring domains [63], adjacent TADs were also found to partially overlap [64], which provides a snippet of information on how AP-2δ targets might be organized in the genome.

In addition, the experimentally validated AP-2δ targets were investigated for their common Transcription co-Factors (TcoFs) [65] in order to identify the co-regulators that might be responsible for mutual genome control orchestrated by AP-2δ. It is worth noting that typically, TFs recruit interactive TcoFs and form physical contact loops [66]. Identification of repeating TcoFs from the TcoFBase server revealed that except for *LINC02126* (which was not included in the database), only 12 TcoFs regulate the rest of the 65 validated AP-2δ targets (see Appendix A for the complete list of TcoFs for each target). These co-factors were: *TLE3*, *TRIM22*, *TRIM24, TRIM25*, *TRIM28*, *TRRAP*, *USP7*, *WDR5*, *XRN2*, *YAP1*, *ZMIZ1*, and *ZMYND8*. Interestingly, *TRIM22* is not only present in the list of currently known AP-2δ targets but also has been experimentally validated [67]. Since the next study stages took a pan-cancer view, CorrelationAnalyzeR was used to correlate *TRIM22* and *TFAP2D* expression in all cancer and normal specimens. Unexpectedly, it appears that depending on the sample type, *TFAP2D* correlates positively (cancer) or negatively (normal) with *TRIM22* (Figure 8D). Together with the fact that *TRIM22* is an AP-2δ target, this suggests a change in the regulation of important TcoFs and thus target genes that are commonly regulated. Although the function might depend on tissue context [68], *TRIM22* was found to manifest oncogenic properties in chronic myeloid leukemia and non-small cell lung cancer [69,70].

### 2.6. TFAP2D-Correlated Genes Regulate Various Processes, including Genome Organization, Whereas AP-2δ Targets Regulate Transcription, as Does Their Superior TF

Subsequently, pan-cancer analysis of AP-2δ was directed to identify genes that correlate with *TFAP2D* in various tumors, and to indicate the most relevant ones both within a specific cancer type (correlating gene is identified for the same cancer in more than one database), and between tumors (correlating gene repeats in various cancer tissues). We concluded that such an approach is more appropriate given the scarcity of cancer research on AP-2δ, and because it might reveal genes that are in fact AP-2δ targets but are not currently considered as such. Moreover, the majority of AP-2δ targets are predicted via Harmonizome and thus not assigned to any specific tissue.

Regarding the within-cancer part, most genes that repeated in the cBioPortal, Gene Expression Profiling Interactive Analysis (GEPIA2), CorrelationAnalyzeR and University of Alabama at Birmingham Cancer (UALCAN) databases were identified as positively correlated (see Appendix A for full report). However, it was also possible to spot negatively correlated genes, as proved by the Uterine Carcinosarcoma (UCS) cohort (Figure 9). Moreover, some genes were found in the list of AP-2δ targets, namely *CDKL3* and *ATP6V0A1* in Adrenocortical Carcinoma (ACC), *VMO1* in Breast invasive Carcinoma (BRCA), *AVP* and *KRT25* in Cholangiocarcinoma (CHOL), *RNF8* in Glioblastoma Multiforme (GBM), *TMEM44* in Lung Squamous Cell Carcinoma (LUSC) and *RPL38* in Sarcoma (SARC). For some cohorts, no *TFAP2D* positively or negatively correlated genes were identified in any database (Appendix A). This applies to Diffuse Large B-cell Lymphoma (DLBC), Kidney Chromophobe (KICH), Ovarian serous Cystadenocarcinoma (OV), Rectum Adenocarcinoma (READ), Stomach Adenocarcinoma (STAD), Skin Cutaneous Melanoma (SKCM), and Uterine Corpus Endometrial Carcinoma (UCEC). In each cohort, separately for positively and negatively correlated genes, the gene ontology was performed using Database for Annotation Visualization and Integrated Discovery (DAVID), revealing a plethora of biological processes and signaling pathways (Appendix A). For instance, functional annotations indicating any genome organization-related processes were identified in four cohorts: Bladder urothelial Carcinoma (BLCA), Liver Hepatocellular Carcinoma (LIHC), CHOL, and LUAD.

In the between-tumors analysis, 1716 *TFAP2D*-correlated genes were identified (Appendix A), including 82 AP-2δ targets that presented both high interconnectivity based on GeneMania (Figure 10A; Appendix A) and implication in, e.g., transcription, cell cycle, pluripotency, neurogenesis, and Vascular Endothelial Growth Factor (VEGF) signaling, based on DAVID (Figure 10B). Nearly 90% of connectivity (considering the network weighting) stems from physical interaction (61.02%) and co-expression (26.38%). The most numerous groups consist of more than 10 annotated genes and are all related to transcription, which allows us to view the subject from a wider perspective. Namely, it is known that spatial genome organization influences the gene expression regulation, but recently it has been also noted that at a finer scale, the transcripts or transcription seem to play a role in sub-compartmentalization, sub-TAD connections, as well as in the stabilization of enhancer–promoter interactions [25]. Knowing that AP-2δ is a protein that regulates transcription of its targets and that these genes further influence transcriptional processes, we hypothesized that AP-2δ might directly or indirectly affect the nuclear organization. In order to verify it, we narrowed the breadth of the study and focused on a suitable tumor for which such investigation would seem justifiable, given the results obtained throughout this research. It turned out that the LUAD cohort is not only in the leading tumors when it comes to *TFAP2D* mutation frequency (Figure 6), but also some cancer hallmark genes and drug targets were found to be differently expressed depending on *TFAP2D* mutation status (Figure 7). Moreover, this cancer was characterized by one of the most numerous groups of *TFAP2D*-correlated genes (Figure 9) and was identified as one of the four tumors (next to BLCA, CHOL, and LIHC) having any functional annotation related to chromatin organization in within-cancer gene ontology (Appendix A). Since the chromatin structure was generally found to change during LUAD progression [71], we certified our decision to continue with this tumor in the remaining study stages.

### 2.7. Insights into LUAD Revealed the Co-Dependence of AP-2δ and Chromosomal Conformation-Related Genes That Present Various Cancer vs. Normal Tissue Staining

The results from the LUAD cohort revealed that both *TFAP2D* expression and mutation status were important for genome organization-related genes; in addition, few examples were repeated in two contexts (Figure 11A). Although these genes were collectively acquired from a single elaboration [72], in the following sentences, the dedicated references confirming the participation of genes in chromosomal conformation are included next to the symbols. Our results present that the patients with a mutation of the AP-2δ–encoding gene have lower expression of *AIFM1* [73], similar to those that have lower *TFAP2D* expression. Contrary results were observed for both *BLM* [74] and *POT1* [75]—lower expression of these genes was found in patients having high *TFAP2D* expression or lacking *TFAP2D* mutation. Other genes for which co-dependence with *TFAP2D* mutation was noted are *CHAF1B* [76] and *HIRA* [77]—both had increased expression in groups bearing *TFAP2D* mutation. The remaining observations concern *TFAP2D* expression; the genes were characterized with either lower (*DHX36* [78], *ERCC3* [79], *GATAD1* [80], *MRE11A* [81], *SMC2* [82], *WRN* [83], *HMGA1* [84], *NBN* [81], *SMC3* [85], *HIST2H3C* [86], *RAD51* [87]) or higher (*ATRX* [88], *PARP10* [89], *TP53* [90]) expression in patients having high *TFAP2D* expression. Comparing the results with the AP-2δ targets list, *HIRA* and *TP53* were identified as genes regulated by this TF. In summary, it seems that LUAD genome organization might differ in groups of different *TFAP2D* expression. The analysis of gene expression changes in relation to lung cancer development revealed some valuable observations. First, high expression of *AIFM1* was found to drive the progression of lung cancer [91], similar to *HMGA1* or *RAD51* overexpression, which was connected with worse LUAD patient outcomes [92,93]. On the other hand, *BLM* or *ATRX* downregulation is associated with favorable events related to Non-Small Cell Lung Cancer (NSCLC) sensitization to radiotherapy or immunotherapy, respectively [94,95]. Finally, decreased expression of either *DHX36* or *ERCC3* seems to be unfavorable due to the elevated NSCLC migration and growth [96] or increased DNA adduct levels [97], even if the latter one was mainly due to *ERCC5* and *ERCC6* (in study by Cheng et al., the mean *ERCC3* expression was lower in oncological patients but the results were not statistically significant). Additionally, the expression of *TFAP2D* in both the wild-type and mutated LUAD samples was compared to normal lung samples (Figure 11B).

The above observations demonstrate the co-dependence of *TFAP2D* expression and genes that might be useful in lung cancer research. Therefore, in the last step, these genes were subjected to further evaluation of their immunohistochemistry (IHC) data (Figure 12). Out of 19 genes, no comparison was possible for *GATAD1* and *POT1* due to a lack of data in both the normal lung tissue and lung adenocarcinoma. For the remaining genes, IHC revealed staining differences for nine genes (*AIFM1, ATRX, DHX36, MRE11, NBN, SMC2, SMC3, TP53, WRN*) and no differences or undetected staining for eight genes (*BLM, CHAF1B, ERCC3, HIST2H3C* (also known as *H3C14*), *HIRA, HMGA1, PARP10, RAD51*). The immunohistochemical data for *TRIM22* was additionally included in this step to complement the results of TcoFs analysis—the gene was found to be expressed more abundantly in LUAD than in normal lungs, which could confirm its oncogenic properties in lung cancer [70]. The other interesting genes are *AIFM1*, *ATRX,* and *DHX36,* which have not only various IHC data, but also the staining complements literature data on LUAD [91,95,96]. In addition, higher cancer-promoting expression is observed in patients with high AP-2δ expression, making *TFAP2D* a potential oncogene of lung adenocarcinoma. Further research is needed in this area, although it is already evident that the lung is the proper direction that can yield valuable data due to the fact that the highest expression of *TFAP2D* is mainly observed in lung carcinoma (Figure 13A; out of ten cancer cell lines with the highest expression of *TFAP2D*, seven stem from lung cancer). Moreover, various AP-2δ expressions can be observed in different lung tumor histological types (Figure 13B; the highest expression in adenocarcinoma), as well as cancer stages (Figure 13C; higher average *TFAP2D* expression in stage I and stage II tumors compared to the normal lung). Thus, when conceptualizing the experimental research, one should consider these important aspects to select cellular models with relevant AP-2δ levels.

## 3. Materials and Methods

### 3.1. Data Acquisition of AP-2 Sequences, Recognized DNA, Evolution, Structures, and Clefts

Sequences of AP-2 proteins were acquired from Universal Protein Resource (UniProt) and aligned on the same server using Clustal Omega v1.2 (default parameters), which also visualized a phylogenetic tree and estimated percentage identity. The consensus logos representing sequences that are recognized by AP-2 factors were obtained from HOCOMOCO v11, except for AP-2ε which is not included in the database. The repository was searched using a “TFAP2” query among human binding models (“full” collection). For all AP-2 factors except for AP-2ε, the positional count matrices (PCMs) were downloaded in order to compare them using the Jaccard similarity index that was estimated through the MAtrix CompaRisOn by Approximate *p*-value Estimation (MACRO-APE) tool, employed with default parameters. The FABRIC database was employed to present the “TF_AP-2” domains and their range. Data on the occurrence of proline-rich sequences in all AP-2 factors were collectively acquired from UniProt, ProSite, InterPro, and Motif Scan.

The three-dimensional structure models in PDB format were acquired from the AlphaFold database and loaded into the ChimeraX v1.4 to visualize similarities and differences in protein structures of all AP-2 factors. PTMs and motifs/regions were identified via the ScanProsite tool with option “1” (submit protein(s); scan motifs). Accession identifiers for all AP-2 members were acquired from the aforementioned UniProt, and the analysis was submitted with an additional option to run the scan at high sensitivity. Histidines in AP-2 protein sequences were highlighted manually on screenshots from AlphaFold. Secondary structures that are formed in AP-2 transcription factors were estimated using the GOR4 algorithm with default parameters. UniProt database was also employed to provide the details about amino acids preceding the conserved “TF_AP-2” domain; the built-in “Highlight properties” option was used to visualize the similarity and various physicochemical or structural properties.

The presence of clefts among AP-2 members was analyzed via PDBsum–UniProt ID provided in the “Alpha Fold model” search option, and data from the “Clefts” tab was then summarized. Moreover, a three-dimensional AP-2δ structure with presentation of the binding site(s) was additionally downloaded from PDBsum in RasMol format in order to present clefts, side by side with AlphaFold models which were intended to visualize the range of the histidine-rich region.

### 3.2. Collection of Data Related to Mutational Status, Genetic Targets, and Transcriptional Co-Factors

Mutation frequency data were collected from FABRIC and The Tumor Immune Estimation Resource (TIMER 2.0). The connection of *TFAP2D* mutation status to gene expression changes was performed via muTarget. Analysis was started with the “Genotype” option; all mutation types and tumor types were included with default threshold (*p* < 0.01 and fold-change ≥ 1.44). Tables and boxplots were generated separately for cancer hallmark genes and FDA-approved drug targets. In addition, the summary of muTarget reports (Appendix A) includes all significant genes which were further subjected to the gene ontology analysis (biological processes) using the built-in option. AP-2δ gene targets were collected from all available sources: databases (Harmonizome which contains, e.g., TRANScription FACtor database [TRANSFAC]); ChIP-exo experiment [61] from Gene Expression Omnibus (GSE151287: currently, to the best of our knowledge, there is only one such experiment); and literature data [67,98,99,100,101]. Collectively, 1318 target genes of AP-2δ were identified (Appendix A).

The genomic location of currently known AP-2δ targets was visualized through Ideogram Viewer v1.0.8, according to official guidelines (i.e., genes symbols were included in the Universal Resource Locator [URL] under the “genelist=” query parameter). For clarity purposes (and to increase the findings relevance), only experimentally validated AP-2δ targets were presented via WashU Epigenome Browser. The hg19 reference genome was selected due to the fact that it allows a three-dimensional visualization of the genome from Public Data Hubs to be loaded under the “Tracks” tab. The 3dg data from the GM12878 cell line [60] were converted by the repository to g3d format using g3dtools. An additional black-gray indicator was drawn to facilitate results interpretation in various orientations (Appendix A). The most interesting observations were manually zoomed and details of genes’ location were retrieved from the left panel of WashU Browser. For *MVK* and *VPS29*, insights into TADs were visualized through TADKB and manually annotated with the help of University of California Santa Cruz Genome Browser (UCSC). For consistency, TADKB was browsed using the same reference genome and cell line (GM12878) as WashU Browser, with 50 kb resolution and the calling TAD method set as “Gaussian Mixture model And Proportion test” (GMAP).

Experimentally-validated AP-2δ targets were also investigated for their common TcoFs using the TcoFBase server. The “search by target gene” option was selected and all possible methods of TcoF identification were employed for Human species. The acquired lists were scanned for duplicates using Kutools v19 to provide TcoFs that regulate all query genes. Only one TcoF was identified as an AP-2δ target, TRIM22; it was further correlated with *TFAP2D* using CorrelationAnalyzeR in all available tissue types at once, but separately for tumor and normal sample types.

### 3.3. Correlation Analysis, Gene Ontology, and Evaluation of AP-2δ Impact on Genome Organization

*TFAP2D*-correlated genes were obtained from cBioPortal, GEPIA2, CorrelationAnalyzeR and UALCAN; separate lists were prepared for positively and negatively correlated genes. A correlation coefficient of at least |0.3| was considered acceptable. After merging, duplicates were first identified using Kutools v19 in data for a specific tumor. After completing the within-cancer part (summarized in circular plot using Cytoscape v3.7 with additional gene ontology via DAVID), duplicates were temporarily removed from data for specific tumors and then merged into a single list to identify duplicates between tumors. These genes were reduced only to AP-2δ targets, which were subsequently subjected to analysis of interconnectivity via GeneMania and gene ontology via DAVID. GeneMania was customized to show zero “resultant” genes (i.e., additional genes that are added to artificially increase the connectivity of query genes) while DAVID was employed under default parameters with the selected identifier set as “OFFICIAL_GENE_SYMBOL”. Only functional annotations that contained at least 3 query genes and presented statistical significance (*p* < 0.05) were collected from DAVID. Only interconnected nodes were subjected to functional annotation.

The assumptions made during the within-cancer and between-tumors parts were evaluated using RNA-seq expression data (level 3 RNA-seqV2, RSEM normalized) of the LUAD cohort from The Cancer Genome Atlas (TCGA). In order to split patients not only based on expression, *TFAP2D*-mutated samples were identified via Genomic Data Commons (GDC) Data Portal. The list of genes implicated in the nuclear organization was retrieved from Labome’s elaboration [72]. Results were visualized using BoxPlotR webtool and manually edited using Inkscape v1.2. Expression differences between groups were evaluated using GraphPad Prism v8; *p* < 0.05 in the Mann–Whitney test was considered statistically significant. Additionally, the representative IHC data were obtained from the publicly-available Human Protein Atlas; the same antibody for both normal and tumor lung specimens was selected using the “Tissue” and “Pathology” atlas, respectively. Selected antibodies had at least an “Approved” validation score, but some of them presented higher scores, e.g., “Supported” or “Enhanced” (Table 2). Expression of *TFAP2D* across cancer cell lines was determined using DepMap (Expression 22Q2 Public source); the portal was searched by gene name, and data were acquired from the “Characterization” tab and Data Explorer tool (groups were colored by the primary disease option). AP-2δ expression in various lung cancer histological types and stages was obtained from Gene Expression patterns across Normal and Tumor tissues (GENT2) and Oncopression, respectively. A “Subtype profile” workflow was selected in GENT2 with the following parameters: “Tissues” set as “Lung”, “Subtypes” as “Histology”, and “Gene Symbol” as “TFAP2D”. Oncopression was queried with the gene symbol, and lung cancer was accessed for further details on staging.

## 4. Conclusions

To conclude, cancer treatment-related research on the AP-2 family of transcription factors is encouraged, especially regarding AP-2δ, given its high ligandability potential, or evolutionary distinction that entails exclusive structural and functional properties. Unequivocal recognition of AP-2δ would require simultaneous identification of the histidine-rich region and “TF_AP-2” domain, in order to exclude other molecules from inside and outside the AP-2 family. The presence of the highest number of PTMs and extended strands confirms the relevance of AP-2δ among family representatives. Moreover, AP-2δ might be a more relevant therapeutic target owing to its presumed interaction with unique coactivators or functioning as a modulator of other AP-2 proteins. The second most interesting AP-2 member could be AP-2γ, for which much literature data confirm its oncogenic properties, and for which our study adds valuable information regarding available clefts.

Various steps of AP-2δ pan-cancer analysis identified different tumors that are relevant to further research. *TFAP2D* mutation may co-depend with changes in expression of both cancer hallmark genes and FDA-approved drug targets. Investigation of experimentally validated AP-2δ targets revealed that their close location in the genome could be due to inter-chromosomal proximity or TADs lying together on a single chromosome. Among TcoFs, the most inquisitive appears to be TRIM22, which regulates all currently validated AP-2δ and its correlation with *TFAP2D* changes depending on tissue context (cancer vs. normal). Based on the literature data, *TRIM22* could manifest oncogenic properties in leukemia or lung cancer. Pan-cancer correlation analysis revealed a plethora of biological processes and signaling pathways that are worth investigating in the future. The between-tumors part of the investigation indicated nearly a hundred AP-2δ targets that in addition to being highly interconnected, also regulate transcription, as does their superior transcription factor. This allowed us to view the subject from a wider perspective, i.e., viewing transcription as a process that shapes the spatial organization of the genome. Assumptions were certified using a representative cohort, LUAD, in which a co-dependence was observed between AP-2δ and the expression of chromosome conformation-related genes; the latter were found to influence lung cancer development and presented various staining in cancer vs. normal comparison. Lung cancer cell lines generally present the highest expression of *TFAP2D*, suggesting that research in this context should be pursued. If genome organization were to be orchestrated by AP-2δ, and if cancer-driving phenomena were found to be implicated in genome organization, targeting this TF may bring tremendous benefits.

Evidently, the results from this study and the scarcity of cancer-related data on AP-2δ strongly emphasize the need for subsequent research aimed at filling both the literature and therapeutic gaps via Cleavage Under Targets and Tagmentation (CUT&Tag) or CUT&RUN (Release Using Nuclease), chromosome conformation capture and CADD.

## Figures and Tables

**Figure 1 cells-11-04124-f001:**
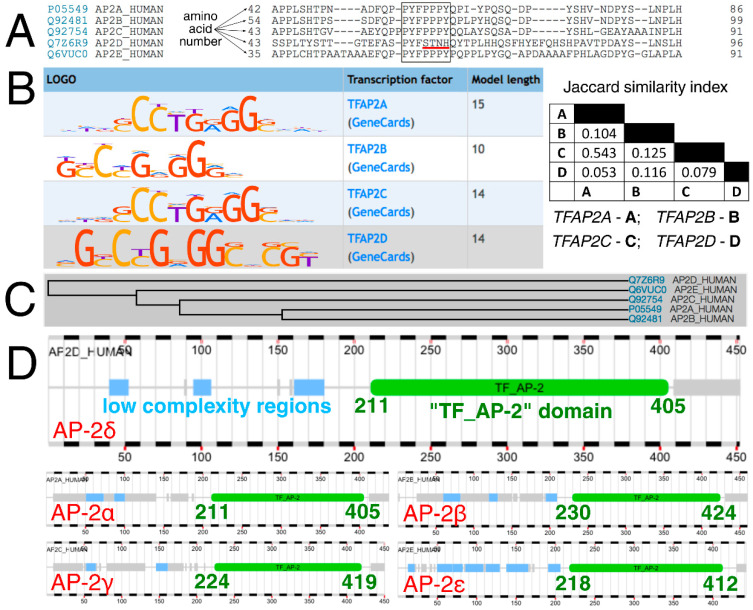
Analysis of AP-2 factors’ sequence, recognizable DNA, evolution, and “TF_AP-2” domain. (**A**) Clustal Omega revealed that the amino acid sequence (encompassing the proline-rich motif in AP-2 members α, β, γ, ε) is different in AP-2δ (the core of the pattern is framed; the main difference in AP-2δ is underlined in red). (**B**) HOCOMOCO identified the consensus logo of AP-2δ to be similar in length to that of AP-2α and AP-2γ, but the sequence itself was congruent to shorter AP-2β according to MACRO-APE. (**C**) Phylogeny data acquired from UniProt proved the separate evolution of AP-2δ compared to other AP-2 proteins. (**D**) The range of the “TF_AP-2” domain among AP-2 members certifies that this part of the protein sequence is highly conserved. This indicates that the distinction of AP-2 proteins is supposedly due to differences located in the “first half” of the polypeptide chain. The “TF_AP-2” domain is marked in green, while the low complexity regions are marked in blue (acquired from FABRIC).

**Figure 2 cells-11-04124-f002:**
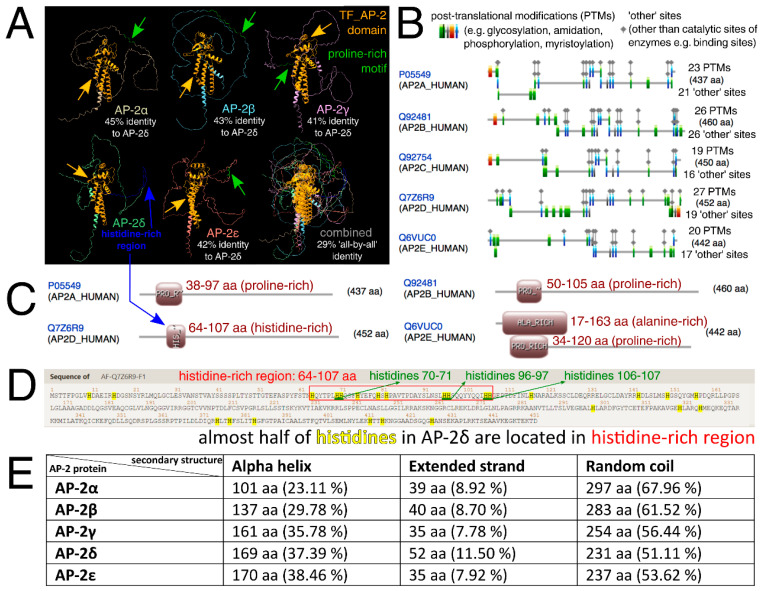
Similarities and differences of AP-2 factors with regard to protein structures, frequent patterns, and amino acid-rich regions. (**A**) Three-dimensional protein structures of AP-2 factors, with additional overlapping, were shown via ChimeraX, while the percentages of structure identity (in relation to AP-2δ) were estimated using Clustal Omega. “TF_AP-2” domains are marked in orange, proline-rich motifs (of AP-2α/β/γ/ε) are in green, while the histidine-rich region (of AP-2δ) is in blue. (**B**) ScanProsite tool revealed that the highest number of PTMs can be found in AP-2δ. (**C**) Based on ProSite data, the histidine-rich region of AP-2δ (shown beforehand in the first subfigure) overlaps with the proline-rich region found in other AP-2 proteins. (**D**) Visualization of AP-2δ sequence with emphasis on histidines (highlighted in yellow) and histidine-rich region (framed in red). Two histidines in a row (“HH”) are underlined in green. (**E**) The GOR4 algorithm (estimating protein secondary structures) indicated that AP-2δ contains the smallest number of random coils but the highest of extended strands. Provided are both the quantity and percentage of amino acids that form these structures.

**Figure 3 cells-11-04124-f003:**
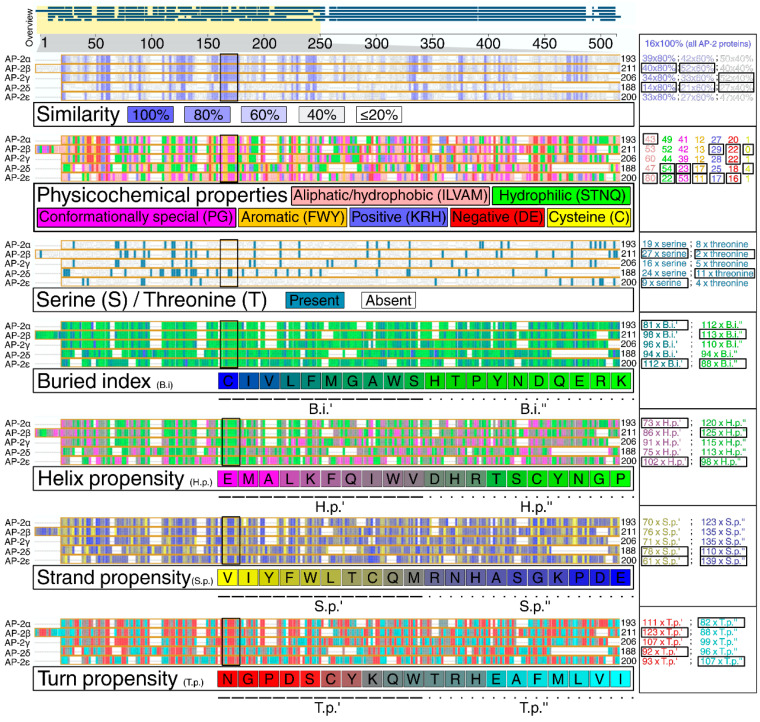
Emphasis on amino acids preceding the conserved “TF_AP-2” domain of AP-2 transcription factors. Data were acquired from the alignment of all AP-2 members using UniProt database. The quantity of amino acids presented for each protein is provided to the right of the sequence, including the far-right panel that subdivides residues based on a specific coloring scheme. Both extremes (the highest and the lowest number among AP-2 proteins) are framed in black. In the middle of the figure, the amino acid sequence encompassing the core of the proline-rich motif of AP-2α/β/γ/ε is also framed in black. “Similarity” indicates the percentage identity of each amino acid between AP-2 proteins. Physicochemical properties are grouped using the Zappo coloring scheme; serines and threonines are also presented in a separate panel. Subsequent panels present structural properties which were dichotomized. Buried index—depending on the frequency of occurrence inside a protein, the amino acid residues are colored from blue (most frequent) to light green (less frequent). Helix propensity—the ability of an amino acid to form an α-helix, colored from magenta (the highest helix propensity) to green (the lowest one). Strand propensity—the ability of an amino acid to form a β-strand, colored from yellow (the highest strand propensity) to blue (the lowest one). Turn propensity—the ability of an amino acid to form a turn, colored from red (the highest turn propensity) to cyan (the lowest one).

**Figure 4 cells-11-04124-f004:**
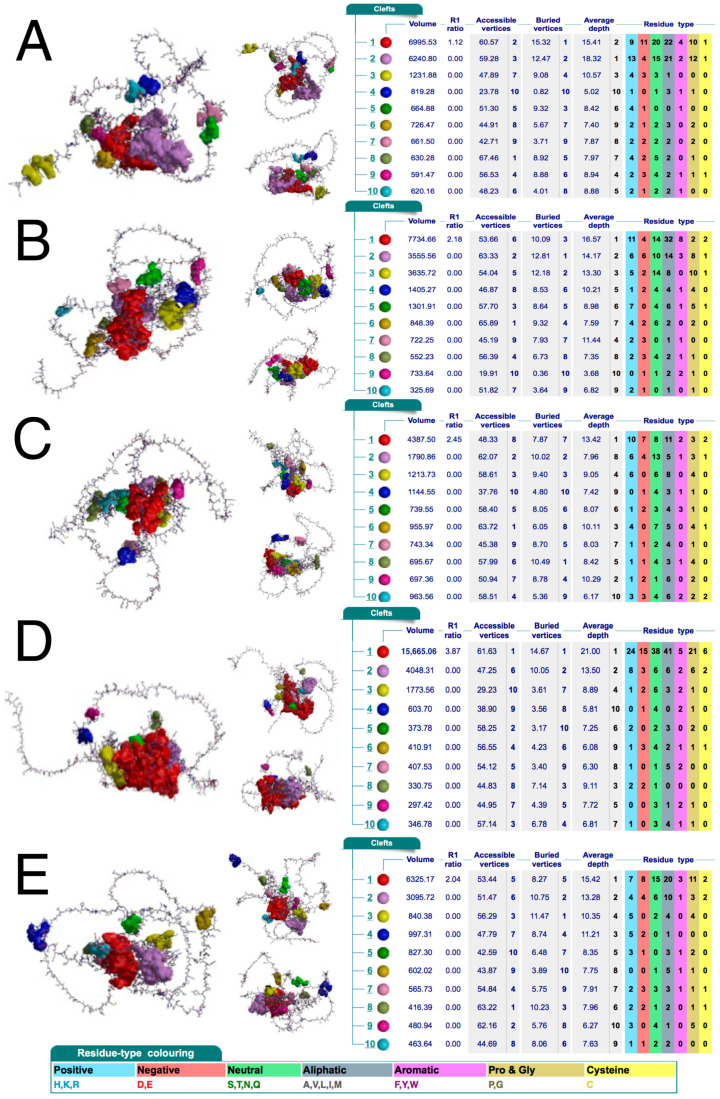
Clefts in the protein structure of AP-2α (**A**), AP-2β (**B**), AP-2γ (**C**), AP-2δ (**D**), and AP-2ε (**E**). Three orientations of protein structure with highlighted clefts (marked in various colors that correspond to those in tables) are shown for each AP-2 transcription factor. Accompanied tables with details on the pocket’s volume, R1 ratio, depth, and residue type are visible on the right; the coloring of the latter is below the last subfigure. Data acquired from PDBsum.

**Figure 5 cells-11-04124-f005:**
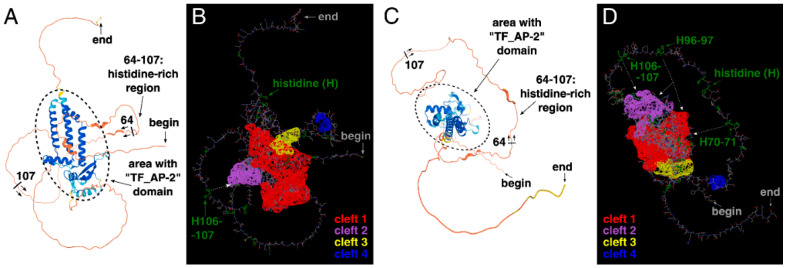
The three-dimensional structure of AP-2δ visualized using AlphaFold and ChimeraX. The emphasis on the histidine-rich region is shown to spot its range, i.e., 64–107 aa; the helices mainly overlapping with the “TF_AP-2” domain are encircled (**A**,**C**). The location of “HH” patterns (two histidines in a row, marked in green) and the largest clefts (highlighted in red, purple, yellow, and blue) are shown (**B**,**D**). Both amino terminus (“begin”) and carboxyl terminus (“end”) of the protein are indicated.

**Figure 6 cells-11-04124-f006:**
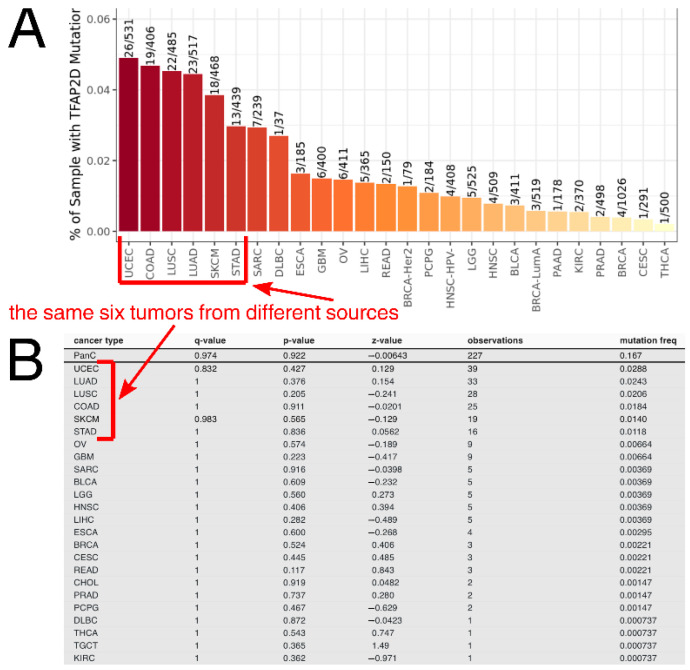
*TFAP2D* mutation frequency across tumors according to TIMER (**A**) and FABRIC (**B**). The same six tumors having the highest mutation frequency (i.e., UCEC, COAD, LUSC, LUAD, SKCM, STAD) are marked with red frames and arrows.

**Figure 7 cells-11-04124-f007:**
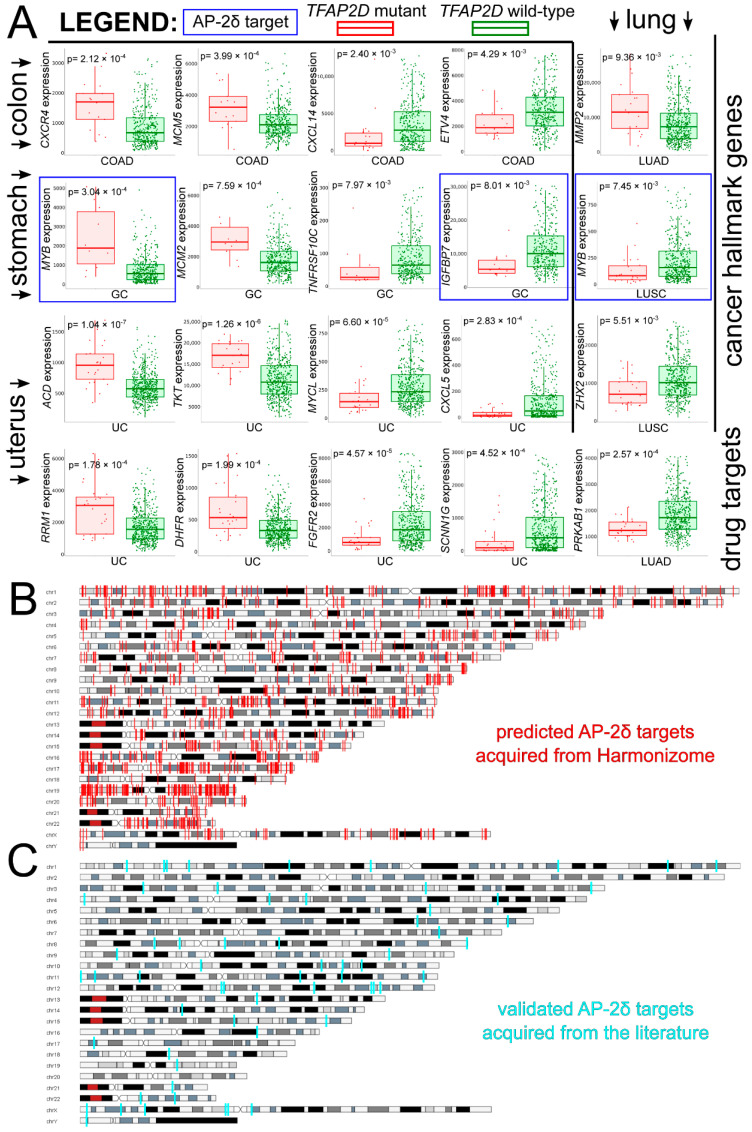
Investigation of AP-2δ mutational status via muTarget and visualization of AP-2δ genetic targets through Ideogram Viewer. (**A**) The best cancer hallmark genes and FDA-approved drug targets of which expression co-dependence occurs with *TFAP2D* mutation status (complete list available in Appendix A). Genes identified as AP-2δ genetic targets are framed in blue. (**B**) Genomic location of predicted AP-2δ targets (marked in red). (**C**) Genomic location of validated AP-2δ targets (marked in cyan).

**Figure 8 cells-11-04124-f008:**
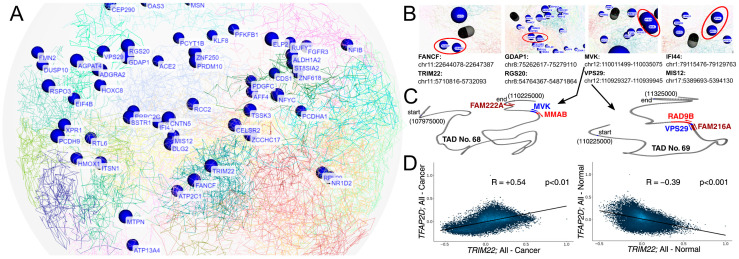
Insights into the three-dimensional genomic location of validated AP-2δ targets (**A**,**B**), TAD domains of *MVK* and *VPS29* (**C**), and correlation of *TFAP2D* with *TRIM22* as the representative of TcoFs (**D**). In subfigure A, the emphasis is initially put on presenting all genes’ names at once. In subfigure B, the most interesting genomic locations were magnified and marked in red. In subfigure C, the locations of *MVK* and *VPS29* (as well as the closest genes) are presented in TADs via TADKB. In subfigure (**D**), the CorrelationAnalyzeR was employed to correlate two genes in all available tissue types, separately for tumor and normal sample types.

**Figure 9 cells-11-04124-f009:**
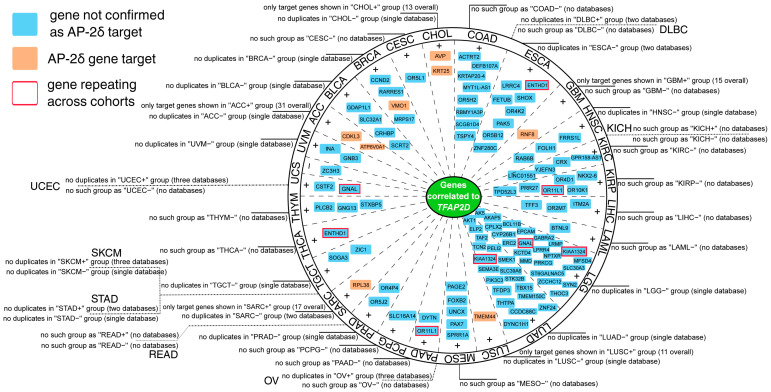
Summary of the most valuable data from within-cancer correlation analysis performed via cBioPortal, GEPIA2, CorrelationAnalyzeR, and UALCAN. Some cohorts revealed no qualifying genes, the cause could be a lack of duplicates across databases (e.g., SKCM), or no results were obtained from any database (e.g., UCEC). Some genes were denoted as AP-2δ targets (orange) or the ones that repeat across cohorts (red frame). Details on data availability are provided around the circle and in Appendix A.

**Figure 10 cells-11-04124-f010:**
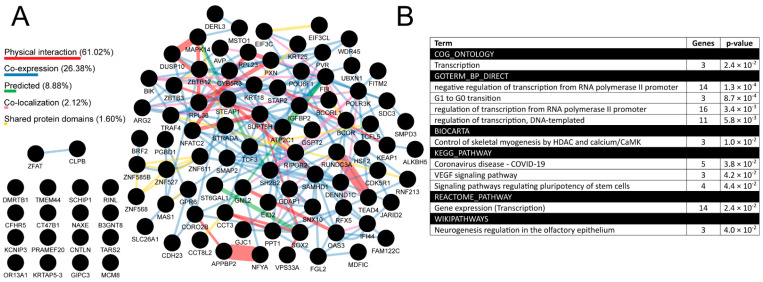
Summary of the most valuable data from between-tumors correlation analysis in the form of interconnectivity (**A**) and gene ontology (**B**). Percentage contribution of a few aspects (marked in various colors) that constitute the interconnectivity of the network is provided. Analysis performed via cBioPortal, GEPIA2, CorrelationAnalyzeR, and UALCAN, visualized using GeneMania and DAVID.

**Figure 11 cells-11-04124-f011:**
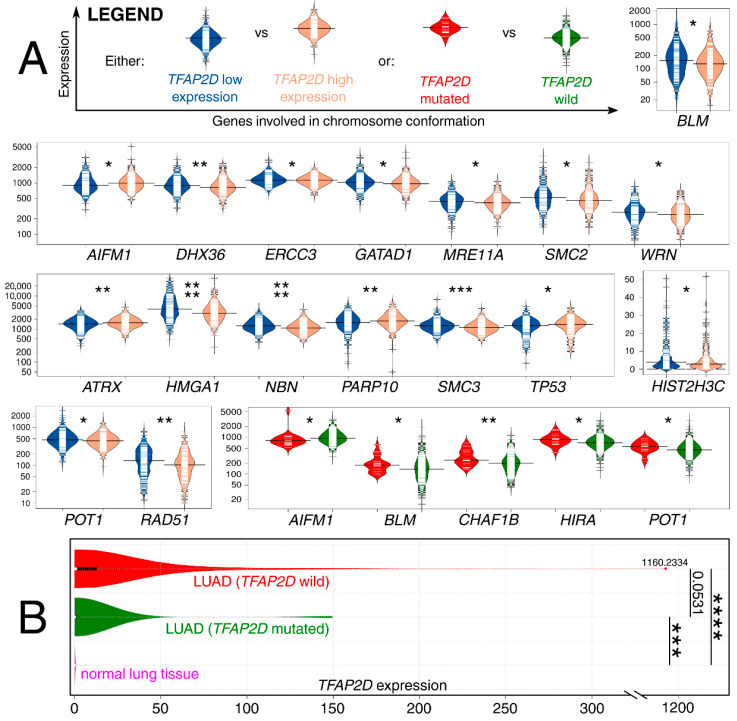
Co-dependence of genes involved in chromosome conformation and either *TFAP2D* mutation or expression. (**A**) Some genes are on separate scales due to large differences in expression levels. (**B**) *TFAP2D* expression was also compared between LUAD (separately for samples with *TFAP2D* wild-type or mutated) and normal lung samples. *p* < 0.05 (*), *p* < 0.01 (**), *p* < 0.001 (***), *p* < 0.0001 (****). Data acquired from The Cancer Genome Atlas and Genomic Data Commons Portal.

**Figure 12 cells-11-04124-f012:**
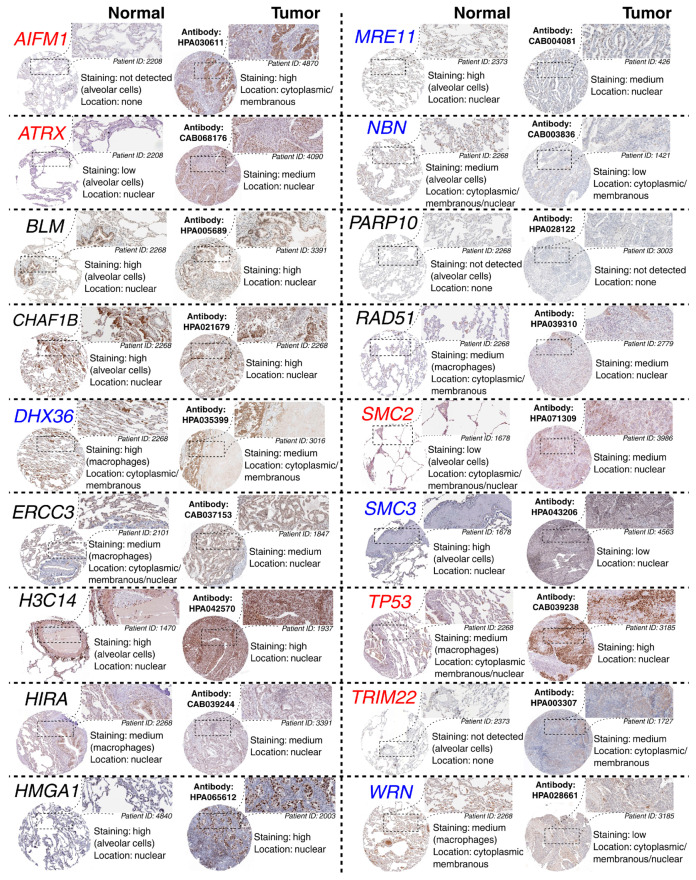
Representative HPA staining data for chromosomal conformation-related genes from normal lung and adenocarcinoma. The genes with various staining in normal vs. tumor comparison are marked in red (if staining is stronger in the tumor) or blue (if staining is weaker in the tumor). One region per specimen is magnified. Antibody catalog number and staining results are provided.

**Figure 13 cells-11-04124-f013:**
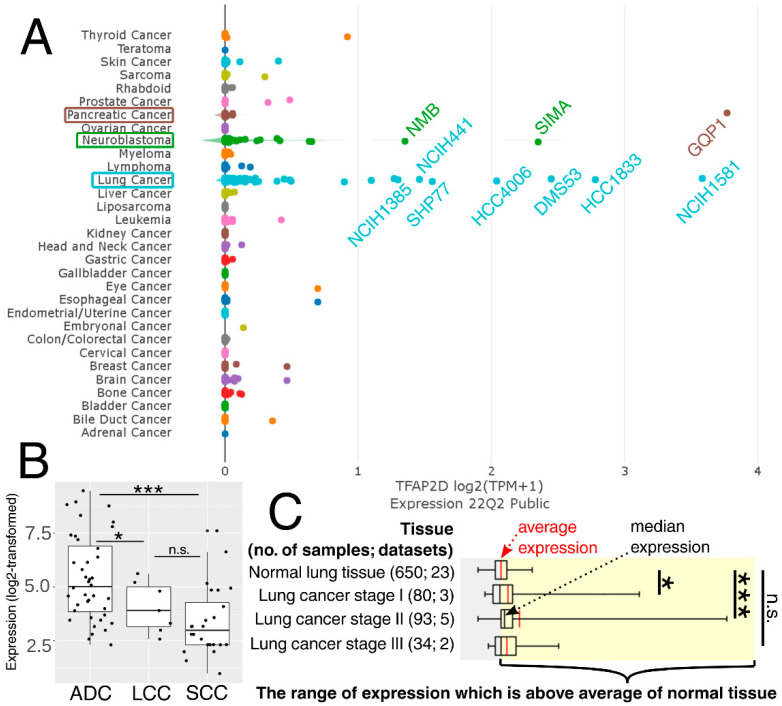
Expression of *TFAP2D* in pan-cancer cell lines (**A**), as well as lung carcinoma histological types (**B**) and stages (**C**). The names of ten cancer cell lines having the highest expression are included. ADC–lung adenocarcinoma. LCC—lung large cell carcinoma. SCC—lung squamous cell carcinoma. *p* > 0.05 (n.s.), *p* < 0.05 (*), *p* < 0.001 (***). Data acquired from DepMap, GENT2, and Oncopression.

**Table 1 cells-11-04124-t001:** Occurrence of proline-rich sequence in AP-2 transcription factors according to databases.

	UniProt	ProSite	InterPro (MobiDB Entry)	Motif Scan
**AP-2α**	57–62 aa ^#^	38–97 aa	49–67 aa	38–97 aa
**AP-2β**	*not available*	50–105 aa	61–79 aa	50–105 aa
**AP-2γ**	59–64 aa	*not available*	*not available*	*not available*
**AP-2δ**	*not available*	*not available*	*not available*	*not available*
**AP-2ε**	54–59 aa	34–120 aa	*not available*	34–120 aa

^#^ aa—amino acids range indicating the proline-rich sequence.

**Table 2 cells-11-04124-t002:** Validation scores of antibodies selected from HPA.

Protein-Encoding Gene	Antibody Catalog Number	Validation Score
** *AIFM1* **	HPA030611	Enhanced–Orthogonal
** *ATRX* **	CAB068176	Supported
** *BLM* **	HPA005689	Approved
** *CHAF1B* **	HPA021679	Approved
** *DHX36* **	HPA035399	Approved
** *ERCC3* **	CAB037153	Approved
***HIST2H3C* (or *H3C14)***	HPA042570	Supported
** *HIRA* **	CAB039244	Supported
** *HMGA1* **	HPA065612	Enhanced–Independent antibodies
** *MRE11* **	CAB004081	Supported
** *NBN* **	CAB003836	Supported
** *PAPR10* **	HPA028122	Approved
** *RAD51* **	HPA039310	Enhanced–Orthogonal
** *SMC2* **	HPA071309	Approved
** *SMC3* **	HPA043206	Approved
** *TP53* **	CAB039238	Enhanced–Orthogonal
** *TRIM22* **	HPA003307	Approved
** *WRN* **	HPA028661	Supported

## Data Availability

The datasets supporting the conclusions of this article are included within the article (and its additional files).

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
