# Peer review of "AP-2δ Is the Most Relevant Target of AP-2 Family-Focused Cancer Therapy and Affects Genome Organization"

_cells, 2022, doi:10.3390/cells11244124_

Round 1
Reviewer 1 Report
In the manuscript “AP-2δ is the most relevant target of AP-2 family-focused cancer therapy and affects genome organization” the authors use numerous in-silico tools to compare AP-2δ to other AP-2 members 2 both from a structural and functional perspectives. They suggest that AP-2δ is therapeutically relevant target in several cancer types and claim that unique PTMs and wider cleft of AP-2δ provide an opportunity for selective targeting.
Overall, I found a large disconnect between most claims and the data presented. The complete lack of experimental and functional evidence and the frail in silico analyses make most claims unsound.
Furthermore, very often the data presented does not support the section headings across the manuscript.
Below are two major concerns of this reviewer:
1 - The claim that AP-2δ affects genome organization, which makes the core of the title too, is bold and unfounded. Correlation between AP-2δ and “Genome organization genes” is far from being sufficient proof to back up such claims. In addition, the authors have to define these genes and their functional relevance to genome organization.
On the other hand, changes in chromatin architecture and organization are not exclusive to LUAD and is a common process in cancer progression and tumorigenesis in general. This can be allocated to changes in cellular states within tumours (i.e plasticity) and/or to sub-clonal evolution, both contributing to most malignant features of cancer.
2 – Suggesting AP-2δ is a pertinent therapeutic target implicates several considerations that authors do not discuss, at least not to the required extent. Members of AP-2 have been heavily involved in developmental and physiological processes. Thus, the relevance of such hypothesis should be discussed from a safety perspective, specially knowing that AP-2δ is more often mutated than amplified.
Specific comments:
Fig1: Describe the tools used to define the DNA binding motif of the different AP-2 members.
Not clear what the authors are trying to say in panel B as all of them fit the AP-2 consensus GCCNNNGGC.
Fig1&2: the authors should leverage the current analyses to highlight the effect of differences AA sequence, in the N-terminal end, on the structure of the different AP-2 members and how does that make AP-2δ more tractable.
Line 215: the authors highlight that AP-2δ is “half identical to other AP-2 factors”. The same is true for all other pairwise comparisons between the other factors. Again, not clear what the authors are trying to say here.
In figure 6a, instead of “cherry-picking” of select genes, the authors should present the data as a global gene expression heatmap to define gene clusters that correlate, or not, with the mutational status of TFAP2D. Then functional analysis (Gene Ontology and pathways) could help define gene networks that could potentially be affected by mutations in AP-2δ. This would be rather correlative than causative, so expansion to multiple cancer types and focusing on common patterns will be helpful.
On the other hand, this emphasizes the importance of defining transcriptional targets AP-2δ (ChIP-Seq or Cut&RUN) – are they different to other AP-2 members? Is the cistrome of AP-2δ different compared to that of the WT?
Once more, it is not clear to this reviewer the purpose of the analyses in Fig 6B and C. It is expected and normal for transcriptional targets of a particular TF to be distributed in different TADs and even distinct chromosomes.
In figure 7: I’d suggest defining TADs by rather integrating analyses of CTCF ChiP-Seq and 3D chromatin looping Hi-C/HiChiP.
To determine potential co-factors of AP-2δ the authors should generate a chip-seq compendium with candidate co-factors and present the data in a unified data matrix to define the degree of overlap in binding genomic regions.
Figure 8 and 9 can be merged in one figure, where the authors could present a two-dimensional gene expression correlation matrix and define clusters of positively/negatively correlated genes with all different members of AP-2. Functional analysis on the identified clusters would also help clarify the gene regulatory networks at play. A stratification based on AP-2δ mutational status is also recommended.
Line 412: the authors should clarify how did they determine that those were AP-2δ targets.
Figure11: in Absence of AP-2δ staining and mutational status, these data seem rather purposeless.
Minor comments:
Add more details to the figure legends.
The quality of certain figures needs to be improved and require a larger font to make them legible (Figure 8a for example) .
Author Response
Dear Reviewer 1, thank you so much for all your comments, please see the attachment for our responses. With kind regards, Damian Kołat

Reviewer 2 Report
The Damian et al analyzed AP-2 family as cancer therapy target, and the AP-2δ affects the cancer genome organization. It is interesting research for targeting the undrggable transcription factors. Moreover, it is important to research the effect on the genome organization in the tumorigenesis. The authors examined the AP-2 family protein mutation, targets, activators, related genes and genome organization impact. The AP-2δ mutations affects various processes of genome organization. The authors conclude that AP-2δ is potential target of cancer therapy of AP-2 family.
The paper is well written, and their results and the implications will be of interest cancer researchers and medicine companies. I think the manuscript should be considered for publication, as long as the authors are able to address some specific concerns (see below).
1, in the manuscript, some figures are blurry for example Figure 7 A B, Figure 8A, Figure 11. It is difficult to read the information in the figures. Could you provide high-resolution figures?
2, For some figure legends, it is too simple to assistant reader understand the research. Could you write them with more detail?
3, Figure 9 A, some of the color of interconnectivity are similar, and it is difficult to read. So please change the color to distinguish each other.
4, The AP-2δ has high potential for ligandability for therapy. Is there molecules available for AP-2δ and potential drugs for the AP-2δ?
5, The co-dependece was observed between AP-2δ and expression of chromosome conformation-related genes. Do you find the direct evidence for AP-2δ affecting the chromosome conformation for example deletion or knockdown the gene effecting on the conformation?
6, In spite of rich analysis of TFAP2D, AP-2δ and other target genes, it still complicated to understand the molecular mechanism of AP-2δ for drug target. Could you make a mechanism figure to summary the whole story of AP-2δ, which is easy for reader to understand?

Author Response
Dear Reviewer 2, thank you so much for all your comments, please see the attachment for our responses. With kind regards, Damian Kołat

Reviewer 3 Report
The authors performed well bioinformatic analysis using publicly available data and indicates that TFAP2D expression and mutation is significant to regulate gene expression associated with genome organization in lung adenocarcinoma. This is an interesting observation.
Major concern
- Authors suggested TFAP2D plays a critical role to regulate gene expression involved in chromosomal conformation in lung adenocarcinoma.
In Figure12, Authors showed TFAP2D expression in cell lines.
Therefore, authors should try siRNA knockdown of TFAP2D in lung cancer cell line expressing TFAP2D to see if TFAP2D regulates chromosomal conformation-related genes in lung cancer cells like author's hypothesis.
Author Response
Dear Reviewer 3, thank you so much for all your comments, please see the attachment for our responses. With kind regards, Damian Kołat

Reviewer 4 Report
The review on the Transcription factor AP-2δ as a potentially important therapeutic target in cancer is timely and well written. A few suggestions are:
1) 1) The authors may want to replace "ChIP sequencing/exonuclease" with newer methods to identify genomic targets of TFs such as Cut & Run and Cut &Tag in the last sentence.
2) The IHC data that is taken from Human Protein Atlas needs to have a disclaimer that the antibodies used in this studies have not been independently validated, at least in some cases.
3) It will be interesting if the authors could add some speculative thoughts about why AP-2δ might be more relevant therapeutic target compared to other family members and whether different stages and types of cancer might be an important consideration.
Author Response
Dear Reviewer 4, thank you so much for all your comments, please see the attachment for our responses. With kind regards, Damian Kołat

Round 2
Reviewer 2 Report
The revised manuscript solved my concerns and in my opinioin it should be accepted.